# Arthritis in Systemic Lupus Erythematosus: From 2022 International GISEA/OEG Symposium

**DOI:** 10.3390/jcm11206016

**Published:** 2022-10-12

**Authors:** Fulvia Ceccarelli, Marcello Govoni, Matteo Piga, Giulia Cassone, Francesco Paolo Cantatore, Giulio Olivieri, Alberto Cauli, Ennio Giulio Favalli, Fabiola Atzeni, Elisa Gremese, Florenzo Iannone, Roberto Caporali, Marco Sebastiani, Gian Franco Ferraccioli, Giovanni Lapadula, Fabrizio Conti

**Affiliations:** 1Lupus Clinic, Dipartimento di Scienze Cliniche Internistiche, Anestesiologiche e Cardiovascolari, Sapienza University of Rome, 00161 Rome, Italy; 2Rheumatology Unit, Department of Medical Sciences, Azienda Ospedaliero-Universitaria S. Anna–Ferrara, University of Ferrara, 44124 Ferrara, Italy; 3Rheumatology Unit, Department of Medical Sciences and Public Health, University Clinic and AOU of Cagliari, 09042 Cagliari, Italy; 4Rheumatology Unit, Department of Surgery, Medicine, Dentistry and Morphological Sciences with Transplant Surgery, Oncology and Regenerative Medicine Relevance, University of Modena and Reggio Emilia, 41121 Modena, Italy; 5Rheumatology Clinic, UOC Reumatologia Universitaria, “Ospedali Riuniti” di Foggia, 71122 Foggia, Italy; 6Division of Clinical Rheumatology, ASST Gaetano Pini-CTO Institute, 20122 Milan, Italy; 7Department of Clinical Sciences and Community Health, Research Center for Adult and Pediatric Rheumatic Diseases, University of Milan, 20122 Milan, Italy; 8Rheumatology Unit, Department of Experimental and Internal Medicine, University of Messina, 98122 Messina, Italy; 9Rheumatology Unit, Fondazione Policlinico Universitario A. Gemelli IRCCS, 00168 Rome, Italy; 10Rheumatology Unit, Department of Emergency Surgery and Organ Transplantations, University of Bari, 70121 Bari, Italy; 11School of Medicine, Università Cattolica del Sacro Cuore, 00168 Rome, Italy

**Keywords:** systemic lupus erythematosus, arthritis, Jaccoud’s arthropathy, rhupus, therapy

## Abstract

Musculoskeletal involvement is one of the most common manifestations of systemic lupus erythematosus (SLE), with a negative impact on both quality of life and overall prognosis. SLE arthritis can be classified into three different subtypes, with different prevalence and characteristic biomarkers and MRI findings. Identifying the pathogenetic mechanisms underlying musculoskeletal manifestations’ development is crucial to develop therapeutic strategies to suppress synovial inflammation, prevent erosions and deformities, and improve SLE patients’ quality of life. Hence, here we discuss the main pathogenetic mechanisms and therapeutic approaches of musculoskeletal manifestations of SLE from the 2022 International GISEA/OEG Symposium.

## 1. Introduction

The “Gruppo Italiano di Studio sulla Early Arthritis” (Italian Group for the Study of Early Arthritis; GISEA) involves 21 hospital and community-based rheumatology units throughout Italy. It has developed and maintained a nationwide registry to promote the study of patients with inflammatory arthritis according to standard-of-care criteria [1]. 

The International GISEA Meeting aims to explore the state of art in many fields of rheumatology, in particular, rheumatoid arthritis, psoriatic arthritis, and axial spondyloarthritis and joint involvement in connective tissue diseases. The present review derives from the 2022 meeting session titled “Joining joints with SLE” and aims to summarize the clinical manifestations, pathogenesis, and treatment of joint involvement in patients affected by systemic lupus erythematosus (SLE).

Musculoskeletal involvement is one of the most common manifestations of SLE, affecting up to 90% of patients [2]. It represents the onset symptom in 60–80% of cases and is reported in up to 60% of disease flares [3,4]. Joint complaints, such as arthralgia or arthritis, are usually transient, but they can mimic rheumatoid arthritis (RA), with persistent pain, swelling, stiffness, and disability. The most frequently affected joints are the metacarpophalangeal and interphalangeal, wrist, and knee, but tenosynovitis or tendonitis could also be present [5].

Lupus arthritis can develop both at onset and during the course of the disease, and it is included in every classification criterion proposed in the last 40 years [6,7]. In the most recent EULAR/ACR classification criteria [8], arthritis is defined as a synovitis affecting two or more joints, characterized by swelling or effusion, or by pain in two or more joints, associated with morning stiffness of at least 30 min.

Musculoskeletal manifestations are a determinant item of SLE disease activity indices. The Systemic Lupus Erythematosus Disease Activity Index 2000 (SLEDAI-2K) is a disease activity score used to stratify the severity of SLE. The presence of arthritis in at least two joints is equal to four points in SLEDAI-2K. However, this evaluation system does not allow for differentiating between an inflammatory articular involvement of only two joints from a more severe one. Further, more than half of patients who have a score = 0 (<2 involved joints) show moderate/high disease activity when evaluated with the Disease Activity Score-28 (DAS28). Of note, DAS28 in SLE patients shares a good correlation with ultrasonographic detection of inflammatory joint involvement as assessed by musculoskeletal ultrasound [9,10].

Joint involvement in SLE could show several clinical phenotypes [11]: (a) joint pain (arthralgia); (b) nondeforming nonerosive (NDNE) or only mildly deforming polyarthritis; more rarely, (c) radiologically nonerosive but deforming arthropathy, namely, Jaccoud’s arthropathy (JA) in 3–13% of patients; and (d) rheumatoid-like erosive arthritis (rhupus: 3–5%) (Figure 1).

Lupus arthritis is usually (80–90%) nondeforming and nonerosive on X-ray, but hands and feet deformities mimicking RA may arise in 5–15% of cases as hallmarks of JA, albeit in the absence of RA-like X-ray bone erosions [12]. Radiographic erosions may be detected in less than 5% of patients showing persistent arthritis, sometimes deformities, high prevalence of rheumatoid factor (RF), and anticyclic citrullinated peptide antibodies (ACPA), thus referred to as rhupus syndrome to indicate an overlap between RA and SLE [13,14,15].

This classification has been recently questioned by ultrasonographic studies that showed an unexpected burden of erosive damage and a surprisingly high prevalence of synovitis (25–90%) and tenosynovitis (25–65%) (see Table 1) [15,16,17,18]. Nowadays, physicians can take advantage of imaging tools that are more sensitive than conventional radiology, such as ultrasound (US) and magnetic resonance (MR). As a result, the boundaries between nonerosive and erosive arthritis in SLE have become decidedly more blurred. In fact, up to 25% and up to 90% of patients (investigated with US and MR, respectively) show erosions of the metacarpophalangeal and proximal interphalangeal joints of the hands and wrists. In recent years, the positivity of ACPA has been observed in a proportion of patients to be between 4% and 27%, and anticarbamylated proteins (anti-CarP) have been documented in 25.6% of patients, while the association with the presence of RF has appeared less evident [19]. MR studies have identified different patterns of lesions in the lupus arthritis subtypes, with NDNE and JA showing prevalent capsular and tendon inflammation, whereas rhupus is characterized by synovial hyperplasia and bone erosions [18,20,21]. Finally, computed tomography (CT) studies of the hands have demonstrated that patients with NDNE and JA have fewer and smaller erosions than that observed in rhupus [22].

The different subtypes of lupus arthritis are also associated with poor health-related quality of life (HR-QoL). For example, patients with active NDNE arthritis show worse HR-QoL impairment measured by the SF-36 questionnaire than those with inactive NDNE arthritis and healthy controls [22]. However, patients with JA and rhupus have the worst scoring compared with those affected by NDNE arthritis in all the HR-QoL domains of the SF-36 questionnaire, including pain and physical function, disability measured by the Health Assessment Questionnaire (HAQ), and fatigue measured by the Functional Assessment of Chronic Illness Therapy (FACIT) score [23]. 

## 2. Pathogenesis of SLE: Focus on the Musculoskeletal Manifestations

Despite the high frequency of lupus arthritis, an understanding of the underlying pathogenic mechanisms remains incomplete. To identify the pathogenetic mechanisms underlying musculoskeletal manifestations’ development is crucial to develop therapeutic strategies to suppress synovial inflammation, prevent erosions and deformities, and improve SLE patients’ quality of life. 

### 2.1. NDNE and JA

Classic synovial histopathology studies from the autopsy of patients with NDNE and JA have revealed scarce inflammatory infiltrate with no or slight synoviocyte hyperplasia, intimal fibrous hyperplasia of blood vessels, and the presence of fibrin and fibrinlike deposits on the synovial layer. Damage was represented by fibrous thickening of the joint capsule and tendon sheaths and loss of cartilage with minimal bone loss (hook erosion) [24,25]. Compared with osteoarthritis and RA, the most recent gene expression analysis from synovial biopsies in SLE patients suggested myeloid cell-driven pathogenesis of lupus arthritis [26]. Evidence of a more robust immune infiltrate than initially reported was found by identifying M1 and M2 macrophages, neutrophils, granulocytes, activated T-cells, and dendritic cell (DC) transcriptomes [26]. Moreover, a significant upregulation of proinflammatory interferon-inducible (IFI) genes and a significant downregulation of genes involved in extracellular matrix (ECM) homeostasis with low tissue repair/destruction activity were displayed [11,26,27]. The type I IFN signature is considered pivotal in maintaining inflammation in established disease, but a clear relationship between IFN activation and disease activity of lupus arthritis has not been established [28]. Type I IFN is a major driver for the B-cell-activating factor (BAFF) production. The BAFF induces B cell activation and differentiation into autoantibody-producing plasma cells with a further generation of the immune complexes able to induce IFN-alpha secretion by plasmacytoid DCs [28]. Moreover, IFN-alpha and BAFF-activated antigen-presenting cells, such as B cells, monocytes, or myeloid DCs, may induce a costimulation to the Th17 subset enhanced in SLE patients, inducing the release of IL-17 [29]. Interestingly, IL-17 and IL-6 are thought to play a pivotal proinflammatory role in lupus arthritis [30,31]. In SLE synovial fluid, the levels of IL-6 correlated with those of IL-17A (r = 0·39, *p* = 0·03, CI = 0·02–0·66) in [31]. IL-6 serum levels also correlated with C-reactive protein serum levels, disease activity, and deformities in patients with SLE in [30]. Cytokines are strongly related to tissue repair and destruction mechanisms, modulating the extracellular matrix (ECM) homeostasis by promoting the production of metalloproteinases (MMPs) and other enzymes. Recently, the serum levels of higher MMP-3 and lower MMP-12 were associated with JA deformities, and most importantly, their levels were associated with capsular swelling and oedematous tenosynovitis, the most prevalent hand MR changes in patients with JA [32]. This evidence suggests that both inflammation and impaired mechanisms of tissue remodeling might be involved in the development of JA.

### 2.2. Erosive Arthritis in SLE

The identification of different phenotypes in SLE-related joint involvement also underlines the need for specific biomarkers able to find patients at risk of developing more aggressive features, including erosive arthritis. Moving from the evidence deriving from RA, in which erosive damage is the hallmark, several studies have evaluated the possible role of RF and ACPA in SLE patients with joint involvement. ACPA are a collection of autoantibodies able to recognize the nonessential amino acid citrulline, resulting from the post-translational modification of arginine residues, catalyzed by peptidylarginine deiminases (PADs) (Figure 2).

The generation of new epitopes, deriving from this process, leads to self-tolerance rupture and to the induction of autoimmune response, with autoantibody production [33]. Data from the literature demonstrated a prevalence for ACPA ranging from 4.4% to 27.3% and for RF from 17.0% to 45.4% [15]. However, only ACPA resulted in being significantly associated with erosive damage, prevalently assessed by conventional X-ray. Furthermore, SLE patients with bone erosions showed significantly higher ACPA titers and polyarthritis compared with patients without bone erosions [34]. Taken together, these data could suggest the possible role of ACPA as a biomarker of SLE-related erosive arthritis. However, the systematic review and meta-analysis performed by Budhram and colleagues demonstrated, despite a high specificity (91.8%), a low sensitivity (47.8%) for ACPA as a biomarker of SLE-related erosive damage [35]. This suggests that a large proportion of SLE patients with erosive damage could be ACPA negative, underlining the need to identify new biomarkers. 

In addition to citrullination, the contribution of carbamylation in RA pathogenesis has been suggested. This is a nonenzymatic post-translational modification, consisting in the addition of a cyanate group on self-proteins. This process leads to the generation of a noncanonical amino acid, the so called homocitrulline (Figure 2). Similar to citrullination, the change in native protein structure leads to tolerance break, resulting in anti-CarP generation [33]. Anti-CarP has been identified in 16% of the so-called seronegative RA, and its presence correlates with more severe erosive damage and with the risk of developing RA in patients with inflammatory arthralgia [36,37,38]. In a study published by Pecani and colleagues, anti-CarP were found in up to 16.8% of SLE patients, irrespective of joint involvement [39]. This prevalence reached more than 40% when enrolling SLE patients with joint involvement; this prevalence was similar to that identified in an RA control cohort [40]. At the same time, Ziegelasch and colleagues assessed two European cohorts from Linköping (Sweden) and Leiden (Netherlands); aCarP were identified in 9.1% of SLE patients, independent from the presence of arthritis. The X-ray assessment, available in 43% of Swedish patients, demonstrated erosive damage in 4.2% of the cohort, but significantly associated with anti-CarP [41]. 

The assessment of 152 SLE patients referring joint involvement, in which erosive damage was investigated by US, confirmed the association between anti-CarP and erosive arthritis. Indeed, a significantly higher prevalence of aCarP was observed in patients with erosions compared with those without. Furthermore, anti-CarP titers positively correlated with US erosive score and with US inflammatory score, suggesting for anti-CarP a possible role as a biomarker of bone damage severity, but also of joint inflammatory status [42]. The contribution of anti-CarP in erosive damage development was confirmed by the application of machine learning models. In this field, anti-CarP antibodies were the most relevant factor determining erosive damage development, followed by ACPA. In detail, anti-CarP contribute 40% to the development of the erosive damage on SLE [42]. 

Next to antibodies, conventionally investigated as biomarkers, other molecules have been evaluated. Among these, the possible role of MMP-3 and MMP-12 serum levels has been investigated, with regard to their ability to discriminate different phenotypes of SLE-related joint involvement. In particular, MMP-3 serum levels were significantly higher in JA patients compared with rhupus syndrome and NDNE arthritis [32]. Finally, the possibility to search biomarkers in other anatomical sites involved in erosive arthritis development from a pathogenic point of view is very attractive. In this context, Sippl and colleagues performed a cytokine analysis of synovial fluid collected from 17 SLE patients. The authors found significantly higher levels of Il-17 and Il-6 in the synovial fluid compared with serum, suggesting a local production for these molecules and, thus, a role exerted in the joint site [31].

## 3. The Modern Treatment of SLE: What about the Musculoskeletal Manifestations

Although arthritis represents one of the most frequent clinical manifestations of SLE, evidence-based treatment options are limited [43,44]. In the therapeutic algorithm proposed by the EULAR recommendations, among conventional disease-modifying antirheumatic drugs (cDMARDs), methotrexate is considered a preferential choice in patients with inadequate control of symptoms after the use of glucocorticoids (GC) and antimalarials at a low/medium dose or in patients in whom hydroxychloroquine alone is not sufficient to control disease activity [43]. In patients with inadequate response to GC and antimalarials, with or without cDMARDs, persistent disease activity prevents a reduction of GC dosage, or in patients with frequent relapses, it is now possible to move towards a second line of treatment with the addition of belimumab [45]. Belimumab is a fully human monoclonal antibody against BAFF. To date, it is the only biotechnological drug (bDMARD) approved for the treatment of SLE. In a phase III placebo-controlled trial, belimumab showed efficacy in controlling joint disease compared with placebo [46]. The efficacy of belimumab was also confirmed in a large multicenter observational study conducted in Italy with 188 patients, among whom joint disease was present in over 45% of the patients. At 12 and 24 months, the presence of arthritis and a high disease activity at baseline (SLEDAI-2K = 10) were independent predictors of response (OR and 32.3 and 12.6, respectively) [47]. 

Based on the pathogenetic mechanisms involved in the synovial environment in patients with lupus arthritis, other therapeutic options are emerging. Among these, the inhibition of IL-17 and IFN-I seems to be the most promising.

Tumor necrosis factors’ inhibitors are not currently recommended for the treatment of SLE [43]. On the other hand, data from small case series or case reports have shown a good efficacy of abatacept, tocilizumab, and rituximab on RA-like arthritis (rhupus) [48,49,50,51,52,53].

Unfortunately, other therapeutic options, such as the inhibition of IL-23 and the use of baricitinib, have not provided sufficiently convincing data, despite the promising results obtained in phase II (ustekinumab) and phase III studies (baricitinib: BRAVE I) [54,55]. Hence, further studies and development programs of these drugs in SLE have been stopped.

Other Janus kinase inhibitors have been tested for the treatment of SLE [56]. Phase I and phase II studies on tofacitinib have focused on cardiometabolic parameters and skin involvement [57], while a phase II RCT is currently ongoing with the aim of evaluating the efficacy and safety of upadacitinib and elsubrutinib in patients with nonrenal lupus, but results are not yet available (NCT03978520).

Recently, the European Medicines Agency (EMA) and the United States of America Food and Drug Administration (FDA) approved a second bDMARD, anifrolumab, for the treatment of moderate/severe SLE [58,59,60]. Anifrolumab is an anti-interferon-α receptor human monoclonal antibody against the IFNARI IFN-I receptor. Two phase III studies (TULIP-1 and TULIP-2) and a phase IIb study (MUSE) provide substantial evidence for the efficacy and safety of anifrolumab for moderately to severely active SLE [61,62]. A recent analysis of pooled data from Tulip studies 1 and 2, focused on the efficacy of the drug on the skin and joint component, showed that anifrolumab obtained a significant 50% improvement in tender joint and swollen joint counts at week 52 in patients with at least six affected joints at baseline [58]. It also showed a good safety profile, without showing an increase in HZV infections in patients on active treatment compared with placebo.

In conclusion, the modern treatment of musculoskeletal manifestations of SLE cannot ignore a careful assessment of the subtype of joint involvement, also using new imaging methods such as US and MR and a more precise patient’s serological profiling. The available international recommendations provide a reasonable road map for the treatment of lupus arthritis in clinical practice. Quickly, the availability of two biological drugs will represent a further step forward to ensure a treat-to-target strategy for the treatment of lupus arthritis [63]. Finally, the recent available data on the pathogenic pathways involved in the development of lupus arthritis will offer further perspectives and new therapeutic scenarios in the near future [64,65].

## Figures and Tables

**Figure 1 jcm-11-06016-f001:**
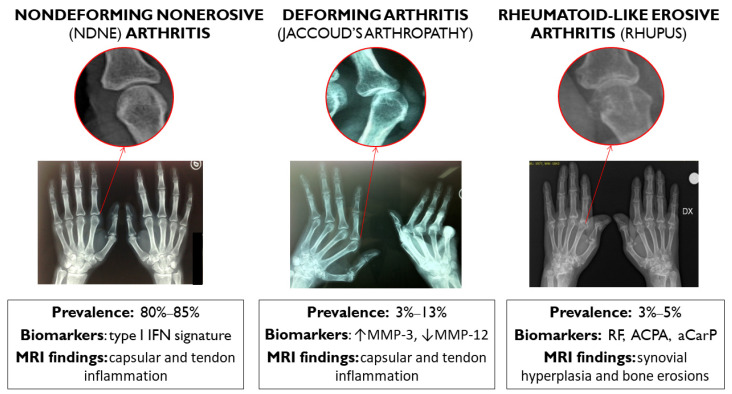
According to clinical findings and traditional X-ray, SLE arthritis can be classified into three different subtypes with different prevalences, biomarkers, and MRI findings: nondeforming nonerosive (NDNE) arthritis; radiologically nonerosive but deforming arthropathy, namely, Jaccoud’s arthropathy; and rheumatoid-like erosive arthritis. IFN: interferon. MMP: matrix metalloproteinase. RF: rheumatoid factor. ACPA: anticyclic citrullinated peptide antibodies. aCarP: anticarbamylated proteins.

**Figure 2 jcm-11-06016-f002:**
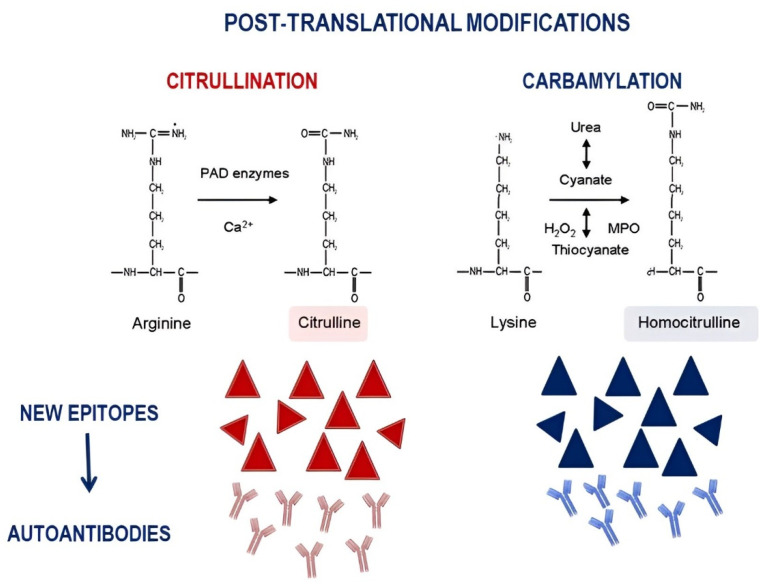
Summary of post-translational modification involved in the production of autoantibodies directed against citrullinated and carbamylated proteins.

**Table 1 jcm-11-06016-t001:** Lupus-related arthritis.

Phenotypes	Prevalence According to Traditional X-ray	Prevalence According to Ultrasound	Biomarkers Available	Treatment
Nondeforming nonerosive arthritis	80–85%	40–60%	Type I interferon signature	First-line treatment: GCs, HCQ, MTX Second-line treatment: belimumab
Deforming arthritis (Jaccoud’s arthropathy)	3–13%	2–47%	MMP3, MMP12	First-line treatment: GCs, HCQ, MTX Second-line treatment: belimumab
Rheumatoid-like erosive arthritis (rhupus)	3–5%	about 5%	RF, ACPA Anti-CarP	First-line treatment: GCs, HCQ, MTX Second-line treatment: belimumab Third-line treatment: rituximab

RCT: randomized controlled trial; MTX: methotrexate; HCQ: hydroxychloroquine; ACPA: anticyclic citrullinated peptide antibodies. aCarP: anticarbamylated proteins.

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
