# Peer review of "Arthritis in Systemic Lupus Erythematosus: From 2022 International GISEA/OEG Symposium"

_jcm, 2022, doi:10.3390/jcm11206016_

Round 1

Reviewer 1 Report

It is a great straight to the point review article. Just as an opinion to help attract more readers, I think the title is a bit misleading, in the sense that for people that do not know what is the GISEA/OEG might just skip the paper while I think if it specifies that is about the study of Arthritis in SLE it might catch the attention better. And I think a table that gather the types of Arthritis/SLE disorders their prevalence, the biomarkers, available treatments and other useful information that authors might consider is of great value to help the general understanding of the article.

Author Response

Response

We thank the referee for the comments.

We modified the title according to their suggestion, replacing in “Arthritis in systemic lupus erythematosus. From “2022 International GISEA/OEG Symposium”

A table has been added

Reviewer 2 Report

This is a comprehensive review on SLE MSK manifestations. I have no major comments for the authors.

This is a comprehensive narrative review article.

Some minor points to consider:

Authors should justify the article's Importance for the readership in the Introduction, as well as one or more concrete aims of the article.

Furthermore, I would suggest the addition of a brief description of the GISEA/OEG Symposium in the Introduction, in order to make it more clear to the reader what information the article will cover.

Typos: Line 33: "characteristic"

Line 34: "findings"

Replace the acronym "LES" with "SLE".

Author Response

We thank the referee for the comments.

We added the aim of the article in the introduction according to the reviewer’ suggestion, other than a brief description of GISEA.

We corrected the typos